



# Extratropical cyclone statistics during the last millennium and the 21st century

Christoph C. Raible[1,2], Martina Messmer[1,2], Flavio Lehner[3], Thomas F. Stocker[1,2], and Richard Blender[4]

[1]Climate and Environmental Physics, University of Bern, Bern, Switzerland
[2]Oeschger Centre for Climate Change Research, Bern, Switzerland
[3]National Center for Atmospheric Research, Boulder (CO), USA
[4]Meteorological Institute, University of Hamburg, Hamburg, Germany

*Correspondence to:* C. C. Raible Climate and Environmental Physics, University of Bern, Sidlerstrasse 5, 3012 Bern, Switzerland raible@climate.unibe.ch

**Abstract.** Extratropical cyclones in winter and their characteristics are investigated in depth for the Atlantic European region, as they are responsible for a significant part of the rainfall and extreme

wind and/or precipitation-induced hazards. Here, we use a seamless transient simulation with a state-of-the-art fully-coupled Earth System Model from 850 to 2100 CE as basis for the analysis. The RCP8.5 scenario is applied in the 21st century. During the Common Era, cyclone characteristics show pronounced variations on interannual and decadal time scales, but no external forcing imprint is found prior to 1850. Thus, variations of extratropical cyclone characteristics are mainly caused by

internal variability of the coupled climate system. When anthropogenic forcing becomes dominant in the 20th century, a decrease of the cyclone occurrences mainly over the Mediterranean and a strong increase of extreme cyclone-related precipitation become detectable. The latter is due to thermodynamics as it follows the Clausius-Clapeyron relation. An important finding, though, is that the relation between temperature and extreme cyclone-related precipitation is not always controlled

by the Clausius-Clapeyron relation, which suggests that dynamical processes can play an important role in generating extreme cyclone-related precipitation - for example in the absence of anomalously warm background conditions. Thus, the importance of dynamical processes, even on decadal time scales, might explain the conundrum that proxy records suggest enhanced occurrence of precipitation extremes during rather cold periods in the past.



## 1 Introduction

Extratropical cyclones are fundamental phenomena of the day-to-day weather variability. Extreme
extratropical cyclones have a strong impact on society and economy and are one of the major natu-
ral hazards of the mid-latitudes (e.g., Schiesser et al., 1997; Beniston, 2007; Etienne et al., 2013).
Thus, a better understanding of variations of cyclone characteristics is essential and has led to a
variety of studies which assess recent and future changes in cyclone characteristics (e.g., Ulbrich
et al., 2008; Bengtsson et al., 2009; Pinto et al., 2009; Raible et al., 2010; Schneidereit et al., 2010;
Zappa et al., 2013; IPCC, 2013). Still, considerable uncertainty remains of how extratropical cy-
clones react to changes of external forcing, especially in the 21st century (Harvey et al., 2012; IPCC,
2013) as confounding and partly canceling processes are difficult to disentangle (O'Gorman, 2010;
Woollings et al., 2012b). Additionally, low-frequency internal variability might be important, e.g.,
a potential influence of the Atlantic Meridional Overturning Circulation (AMOC) on cyclones has
been discussed (Woollings et al., 2012a, 2015). Some of the uncertainties also arise from the fact
that the observed time period is rather short, making it difficult to validate model-simulated decadal
variability in cyclone statistics. Further, there are only a few modelling studies which put changes of
extratropical cyclone characteristics in a long-term perspective (Fischer-Bruns et al., 2005; Raible
et al., 2007; Gagen et al., 2016). A possibility to overcome this is the last millennium which enables
us to study the external forcing imprint on extratropical cyclone characteristics and the interaction
of internal variability of the climate system with these characteristics (Bothe et al., 2015; Smerdon
et al., 2017).

The purpose of this study is to establish a long-term, pre-instrumental perspective for cyclone char-
acteristics. In particular, we evaluate the future of these characteristics under the RCP8.5 scenario
and compare it to natural variability during the last millennium. We take advantage of a transient
simulation for the last millennium in high resolution (approx. $1° \times 1°$) which provides 12-h output —
a necessity to investigate extratropical cyclones and their characteristics, such as cyclone-associated
wind and precipitation extremes. The focus of the analysis is on the North Atlantic region and winter
(December to February).

Various studies have analyzed the climate change response of extratropical cyclones and some of
their characteristics (e.g., Ulbrich et al., 2008; Zappa et al., 2013; IPCC, 2013). A robust finding is
that the warmer atmosphere in the future leads to a moistening of extratropical cyclones and thus to
more precipitation (Bengtsson et al., 2009; Zappa et al., 2013). In the North Atlantic, an extension
of the storm track into Europe is suggested under future climate change (e.g., Bengtsson et al.,
2006; Catto et al., 2011; McDonald, 2011; Zappa et al., 2013). Further modelling studies suggest
that the wind intensity of extratropical cyclones in the North Atlantic is projected to be enhanced
in the future compared to today, leading to a higher potential of future losses (Pinto et al., 2012).
A process relevant for this intensification is a local minimum in the warming of the North Atlantic



Ocean due to a reduction of the AMOC (Rahmstorf et al., 2015). This temperature anomaly leads to stronger temperature gradients within the North Atlantic basin than today and thus to enhanced low-level baroclinicity (Laine et al., 2009; Catto et al., 2011; Woollings et al., 2012b).

Substantial uncertainty remains in future projections of extratropical cyclone characteristics because
of the processes involved (Harvey et al., 2012). The decrease of the projected meridional temperature gradient on average (due to polar amplification) implies a decrease of storm activity in the future, but at the same time the vertical temperature gradient decreases over the Atlantic and Arctic, which induces a reduced static stability and thus a favoring of storm growth (Harvey et al., 2012). Additionally, the moisture changes also influence the cyclone formation as latent heating arising from moist
condensation often strengthens cyclones due to diabatic potential vorticity anomalies (e.g., Gutowski et al., 1992; Li et al., 2014), such that one would expect an intensification of cyclones in a warmer, moister climate (Willison et al., 2013). However, increased moisture, and therefore latent heat content in the global circulation leads to a more efficient poleward transport of energy, and therefore to a weakening of cyclonic activity in the mid-latitudes (e.g., O'Gorman and Schneider, 2008; Schneider
et al., 2010; Li et al., 2014). Additionally, other processes like changes in the wave-wave interaction (James and James, 1989; Riviere, 2011) and in the eddy length scale (Kidston et al., 2011) might play a role for the response of extratropical cyclones to future anthropogenic forcing changes and for the uncertainty of the response in different climate model simulations.

The past can serve as a test bed to place future projection of extratropical cyclone characteristics
into context and to assess multi-decadal variability. Climate states completely different from the present, like the Last Glacial Maximum show pronounced differences in extratropical cyclone behavior. Hofer et al. (2012a,b) showed that cyclones tend to move more zonally over the North Atlantic leading to enhanced precipitation in Southern Europe in winter. The reason is a southward shift of the eddy driven jet due to the Laurentide ice sheet (Merz et al., 2015). However, in the last
interglacial, the Eemian (130 ka ago), the jet positions and thus the cyclones are similar to present (Merz et al., 2015). More relevant is potentially the recent past, i.e., the last millennium including the Medieval Warm Period (approx. 11th to the 13th century) and the Little Ice Age (LIA, approx. 14th to the 19th century; e.g., Bradley and Jones, 1993; Broecker, 2000; McGregor et al., 2015), as these periods are precursors of the Anthropocene (Zalasiewicz et al., 2010) and, thus, provide a rich
and highly resolved proxy network (e.g., PAGES 2k Consortium , 2013). In multi-century preindustrial climate model simulations, Fischer-Bruns et al. (2005) suggested that natural variability of extratropical cyclones is unrelated to external forcing like total solar irradiance (TSI), or volcanoes. This is in contrast to the modelling study of Raible et al. (2007) who found a significant intensification of cyclones in the North Atlantic during the Maunder Minimum (a period of reduced TSI
from AD 1640-1715) compared to today, although part of the signal is already of anthropogenic origin. They further showed that low-level baroclinicity is enhanced due to the increased meridional temperature gradient, which seems to be the dominant process for cyclone intensification in their



coarsely resolved simulations. Comparing this model result with proxy records shows that during the LIA more severe storms are observed (Björck and Clemmensen, 2004; de Jong et al., 2007; Sabatier et al., 2012; Trouet et al., 2012; Van Vliet-Lanoe et al., 2014; Degeai et al., 2015; Costas et al., 2016).

Since the early attempts to assess past extratropical cyclone behavior in model simulations (e.g., Fischer-Bruns et al., 2005; Raible et al., 2007) the ability to perform millennium-size simulations in high resolution has improved so that today several simulations for the last millennium based on different models are available (Schmidt et al., 2011; Braconnot et al., 2012; Taylor et al., 2012; Otto-Bliesner et al., 2016). Still, most of these simulations have only saved monthly data, which prevent us to analyze extratropical cyclones in these simulations. Recently, a last millennium simulation spanning the period AD 850 to 2099 became available providing 12-h data (Lehner et al., 2015). This enables us to address the following research questions:

– How are cyclone characteristics projected to change in the 21st century?

– How do these changes compare with variability, in particular low-frequency variation, during the last millennium?

The study is structured as follows: Section 2 briefly presents the model and experimental design chosen to generate the last millennium simulation. Further, the cyclone detection and tracking method is introduced and the cyclone characteristics are defined. In Section 3 the last millennium simulation is compared with ERA interim for the period AD 1980 to 2009 to demonstrate the model's ability in simulating cyclone characteristics. Then, the climate change signals of the different characteristics are put into context to the low-frequency variability (Section 4). Finally, the results are summarized and discussed and concluding remarks are presented in Section 5.

## 2 Model and methods

### 2.1 Model and experimental design

To investigate the characteristics of extratropical cyclones we use the Community Earth System Model (CESM, 1.0.1 release; Hurrell et al., 2013). It is a state-of-the-art fully-coupled Earth System Model developed by the National Center for Atmospheric Research. CESM relies on the Community Climate System Model (CCSM; Gent et al., 2011) in terms of the model physics, but it contains a carbon cycle module, which is included in its atmosphere, land and ocean components.

The CESM is used in the so-called 1° version to simulate the entire last millennium from AD 850 to 2099. The finite volume core of the atmosphere has a uniform horizontal resolution of $1.25° \times 0.9°$ at 26 vertical levels. The initial conditions for this transient simulation are obtained from a 500-yr control simulation for perpetual AD 850 conditions, which was run into a quasi-equilibrium state





(no drift of the global mean temperature in the upper part of the ocean). The transient external
forcing follows the Paleo Model Intercomparison Project 3 (PMIP3) protocols (Schmidt et al., 2011)
and the Coupled Model Intercomparison Project 5 (Taylor et al., 2012). It consists of TSI, volcanic
and anthropogenic aerosols, land use change, and greenhouse gases (GHGs; Fig. 1). Note that the
TSI deviates from the PMIP3 protocol in that the amplitude between the Maunder Minimum (1640-
1715) and today is doubled. Further, the model has enabled the carbon cycle module. To extend
the simulation beyond AD 2005 the RCP8.5 is applied, which corresponds to a radiative forcing of
approximately 8.5 W m$^{-2}$ by 2100. Further details on the simulation are summarized in Lehner
et al. (2015).

The analysis is based on 12-h instantaneous output, a resolution sufficient to derive characteristics
of extratropical cyclones. The analysis focuses on the North Atlantic region in winter (December to
February, DJF).

### 2.2   Cyclone detection, tracking, and characteristics

The cyclone analysis is based on a modified Lagrangian cyclone detection and tracking scheme first
developed by Blender et al. (1997). The method is applied to the 1000-hPa geopotential height field
and consists of two steps: (i) cyclone detection and (ii) tracking:

(i) A low pressure system is identified as a minimum in the geopotential height at 1000 hPa in a
neighborhood of eight grid points and its intensity (in gpm/1000 km) defined as the mean gradient
between the local geopotential height minimum and its neighboring grid points within an area of
1000 km distance to the minimum. To neglect weak or unrealistic minima, a minimum threshold
value of this intensity measure is set to 20 gpm/1000 km. Further, cyclone centers identified in high
topography (above 1000 m a.s.l.) are excluded.

(ii) To connect identified pressure minima a next neighborhood search is applied within a search
radius of 1000 km. To further prevent erroneous detection of cyclones, two additional thresholds are
used: the cyclone has a minimum lifetime of 24 hours, and the intensity (defined above) needs to
exceed 30 gpm/1000 km once in its lifetime. More details on the cyclone detection algorithm are
provided in Blender et al. (1997), Raible and Blender (2004) and Raible (2007). Furthermore, an
intercomparison of different cyclone detection and tracking methods showed that the method used in
this study is within the range of other methods (Raible et al., 2008; Neu et al., 2013). In particular,
the agreement between the methods increases when focusing on extreme cyclones (Neu et al., 2013;
Lionello et al., 2016; Grieger et al., 2018).

The Lagrangian cyclone detection and tracking method provides a variety of extratropical cyclone
characteristics. Besides the number of time steps when a cyclone is present (or cumulative cyclone
presence), intensity measures for wind are deduced, i.e., the central sea level pressure and the cyclone
depth. The latter is the difference between the central geopotential height and the surrounding mean





geopotential height in distance of the radius of the cyclone (defined below). The 90th percentile of central pressure and cyclone depth of all cyclones within a season is used to define extremeness of wind related measures.

The radius is estimated by a Gaussian radius-depth method based on Schneidereit et al. (2010) to estimate the geometric structure of the cyclones. Thereby, the geopotential height surface in the neighborhood of a cyclone minimum is approximated by a Gaussian, which is fitted by a least squares method. The standard deviation of the Gaussian is then an estimation of the cyclone radius, another characteristic of extratropical cyclones.

Further, the area of the cyclone defined by this radius is used to quantify the amount of precipitation, related to this cyclone. The precipitation is integrated over this area for each time step of the cyclone and defines the cyclone-specific precipitation. To focus on extreme cyclone-specific precipitation, the 90th percentile of cyclone-related precipitation estimated within the season is used as an index of extremeness. A similar approach is selected to deduce cyclone-related temperature, though for this index we are interested in the mean of the season and not in the 90th percentile.

The radius is also used to deduce the Eulerian measure of cyclone occurrences, the so-called cyclone frequency. For one time step each grid point within the radius of a cyclone is assigned to be occupied by the cyclone. Summing over all time steps for each grid point and dividing by the total number of time steps results in cyclone frequency at each grid point. This measure enables us to identify regions of high and low cyclone occurrence.

All extratropical cyclone characteristics mentioned above are deduced for the North Atlantic region defined as $30° - 70°$N and $65°$W $- 40°$E (Fig. 2a).

## 3  Model evaluation

Before extratropical cyclone characteristics for the last millennium and the future are presented, the model's ability to simulate cyclones is demonstrated for the period CE 1980–2009 for winter (DJF). To compare the simulated cyclones and their characteristics the ERA interim reanalysis data are used (Dee et al., 2011). The ERA interim data are first bi-linearly interpolated to the same resolution as CESM ($1.25° \times 0.9°$).

Figure 2 shows the cyclone frequency of the CESM simulation and ERA interim. The main centers of enhanced cyclone occurrence are realistically simulated, i.e., cyclone genesis region over Northern America, the North Atlantic storm track, as well as the Island low pressure region. Still, the CESM tends to simulate more cyclones over the North Atlantic compared to ERA interim. Some differences are found around Greenland and the Hudson Bay where CESM overestimates the cyclone frequency. The reason for this is partly the fact that geopotential height over orography is extrapolated to 1000 hPa leading to artificial high pressure and thus a tendency to weak low pres-



sure systems in the surrounding ocean regions. Still, most of the cyclones in the Labrador Sea are cyclones originating from the cyclone genesis area around Newfoundland. Another caveat is visible over the Mediterranean where CESM slightly underestimates cyclones over the western and central Mediterranean and overestimates cyclone occurrence in the eastern part. Thus, the interpretation of the results over polar regions around Greenland and the Mediterranean requires particular caution.

To further assess the model's ability in extratropical cyclone simulation, distributions of different cyclone characteristics are presented in Fig. 3. We focus on the area North Atlantic marked in Fig. 2a. In total 12369 cyclones are identified in CESM and 7624 in ERA interim. The life time of the cyclones show a similar distribution for CESM and ERA interim (Fig. 3a). CESM shows a slight overestimation of short-lived cyclones and underestimates long-lived cyclones ($> 48$ h). This

is a first hint that more weak cyclones are identified in CESM compared to ERA interim explaining partly the higher cyclone frequencies (Fig. 2). The results of the radius confirm this finding as CESM tends to underestimate the radius, still the shape of the distribution agrees with ERA interim. The wind-sensitive measures show an interesting behavior. Although the shape agrees between CESM and ERA interim, CESM shows more wind intensive cyclones when considering the measure cy-

clone depth. The central SLP measure shows a similar behavior with lower central SLP for CESM than for ERA interim, but also the cumulative cyclone presence with high central SLP is increased in CESM compared to ERA interim. Again the latter indicates that weak cyclones lead to higher cyclone frequencies (Fig. 2). The cyclone-related precipitation shows that CESM slightly underestimates precipitation except for extreme precipitation events ($> 17$ mm/day).

Besides the distributions, the model should also be able to simulate interannual connections between the indices (if they exist). To uncover such connections, the cumulative cyclone presence, median of the cyclone radius and the 90th percentile of cyclone depth, SLP and cyclone-related precipitation in each winter season are estimated and the resulting time series are considered. Table 1 summarizes pairwise correlations for these quantities derived from CESM and ERA interim, respectively.

Significant correlations (5 % significance level) are found between the two wind-related intensity measures, extreme cyclone depth and SLP as well as with the cumulative cyclone presence and these two measures. Most of the significant observed correlations are reproduced by CESM though with slightly lower coefficients. The observed correlation between cumulative cyclone presence and the median radius is not simulated by CESM, while cyclone depth and radius are correlated in CESM

but not in the observations.

In summary, CESM is able to realistically simulate cyclones, their extreme characteristics and the connections among different cyclone characteristics. Some of the discrepancies from ERA interim can be traced back to the tendency that CESM overestimates the number of weak cyclones.



## 4 Results

### 230    4.1   Cyclone intensities during the last millennium

To investigate periods of different cyclonic activity, we define moving averaged indices for all cyclone characteristics, i.e., the cumulative cyclone presence, median of the cyclone radius and the 90th percentile of cyclone depth, SLP, and cyclone-related precipitation (definition see Section 2). First, the indices are estimated for each winter season separately and then averaged over 30-year

periods. The resulting time series are shown in Fig. 4.

In the years previous to AD 1850, all indices exhibit strong decadal to multi-decadal variability (Fig. 4). The cumulative cyclone presence shows a clear negative trend after AD 1850 and leaves the preindustrial range around the year 2000. Thus, the CESM projects a lower number of cyclones for the 21st century in the North Atlantic. The geometry illustrated by the median radius of the cyclones

remains unaffected by external forcing at a first glance and varies between about 215 to 222 km.

Extreme cyclone depth and central SLP both show a weak trend (positive for cyclone depth and negative for central SLP, but not significant) over the entire time series towards higher wind extremes by the end of the 21st century. They also agree in some of the periods with high intensities, e.g., decades around the years 1300s, 1400s, 1680s, 2060s but around year 2000 extreme SLP indicates its lowest

values whereas extreme cyclone depth seems to indicate average years. This difference is a clear indication that it is useful to investigate different intensity measures to conclude on cyclone-related wind extremes. Note that cyclone depth is more related to wind due to the geostrophic approximation compared to SLP which might be influenced by the background pressure (if more cyclones are detected in the low pressure belt they will have deeper central pressure but not necessarily stronger

winds).

Extremes in cyclone-related precipitation clearly react to external forcing. Already before 1850, colder periods (17th century and 19th century) which are partly caused by reduced solar and enhanced volcanic forcing show lower than average 90th percentile cyclone-related precipitation whereas warmer periods are associated with higher than average values. Clearly, the warmest period in the

simulation is the 21st century and there a strong and significant positive trend is simulated.

Some of the new results presented here confirm earlier studies with coarser resolved coupled climate models, e.g., the decrease of cyclone time steps from the preindustrial to the future climate state (e.g., Raible et al., 2007). Still, there are also differences. Raible et al. (2007) suggested a decrease in wind-related intensity from the Maunder Minimum to the present day climate state and

attributed this decrease to generally reduced baroclinicity. This is in contrast to the new simulation where no clear sign of an intensification or weakening is found. A major difference between the two analyses is the resolution of the model (in this study around 1°, in Raible et al. (2007) around 4°). Given the high internal variability, as illustrated by the decadal to multi-decadal variations of the two



wind-related indices, the different processes responsible for extreme cyclones (like baroclinicity in

the lower and upper troposphere, meridional temperature gradient, diabatic processes) may interplay differently in the new simulation. At least diabatic processes are better resolved in CESM compared to the earlier study.

### 4.2   Natural forcing impact on cyclone characteristics

So far there is no clear sign that natural external forcing (volcanoes and solar variations) has a strong

influence on cyclone characteristics whereas at least cyclone-related precipitation shows a strong trend during the period of strong anthropogenic forcing. To disentangle natural and anthropogenic forcing impacts we first focus on the potential volcanic and solar influence during the period 850-1850 CE.

To illustrate the volcanic forcing impact the superposed epoch analysis is applied to the extratropical

cyclone characteristics. The 10 strongest volcanic eruptions, according to optical depth anomaly, over the period 850-1850 CE are composed and time series of the different cyclone characteristics are presented as deseasonalized monthly anomalies from the 5 years preceding an eruption (similar to Lehner et al. (2015)). None of the cyclone characteristics show a volcanic forcing influence (therefore not shown). In particular, wind-related and precipitation-related extremes show no reac-

tion after strong volcanic eruptions although the North Atlantic Oscillation tends to be in its positive phase (Ortega et al., 2015) which has been suggested to be related to wind intense extratropical cyclones (Pinto et al., 2009). The missing volcanic forcing impact on precipitation-related extremes seems to be unexpected as global mean precipitation shows a clear reduction after strong volcanic eruptions (e.g., Frölicher et al., 2011; Muthers et al., 2014; Lehner et al., 2015). Thus, the results

suggest that extremes in both wind and precipitation seems to be decoupled from the mean behavior.

A potential connection between cyclone characteristics and solar variations is investigated by correlating the 30-yr running mean time series (Fig. 4) with the solar forcing (Fig. 1) over the period 850-1850 CE. The analysis with all cyclone characteristics shows that none of the characteristics have a significant correlation with the solar forcing (the highest correlation coefficient is 0.19 be-

tween solar forcing and extreme central pressure). We also tested lag correlations of up to $\pm$ 30 years, but again the correlations were not significant at the 5 % level. Thus, a linear connection of mean and extreme cyclone characteristics to solar forcing is not found in the pre-industrial period of this simulation.

### 4.3   Low-frequency variations of extreme cyclone characteristics during the last millennium

In the following we will focus on the analysis of the two extreme cyclone characteristics: 90th percentile of cyclone depth and of cyclone-related precipitation over the North Atlantic region. To obtain information on the low-frequency of extreme cyclone characteristics, 30-years running aver-





aged periods are investigated in more detail.

Correlation patterns between extreme cyclone depth with different variables like, 2-m temperatures,
500-hPa geopotential height, cyclone frequency and mean precipitation show distinct significant
(5 % level) patterns in the North Atlantic and over Europe (Fig. 5). Low-frequency variations of
extreme cyclone depth correlate negatively with 2-m temperatures around Greenland and positively
over Northern and Eastern Europe (Fig. 5a). This correlation is consistent with the correlation
found between extreme cyclone depth and 500-hPa geopotential height (Fig. 5b) which resemble
a NAO-like structure, but slightly shifted to the north-east, in particular the center located over the
Mediterranean Sea. The center north of Iceland is baroclinic, as the corresponding centers of the cor-
relation patterns with the 1000-hPa geopotential height are shifted to the east resulting in a westward
tilt with height (not shown). The negative correlations of the 2-m temperature around Greenland
go hand in hand with negative correlations between extreme cyclone depth and the sea surface tem-
perature (SST, not shown). This reduction is present over the entire North Atlantic basin, even if
the southern part of the Atlantic does not show a statistically significant change. Furthermore, these
negative correlations co-occur with positive correlations with sea ice around Iceland (not shown).
Additionally, an increase in extreme cyclone depth is related to reduced cyclone frequency and
to a reduction of cyclone-related precipitation around Greenland and an increase in both measures
around Scandinavia (Fig. 5c,d). Thus, the negative geopotential height anomaly (enhanced low pres-
sure system in the mid of the atmosphere) steers the track of cyclones towards Scandinavia where
the cyclone frequency correlates positively with extreme cyclone depth. Furthermore, the correlation
pattern of extreme cyclone depth with 2-m temperature show that under increased extreme cyclone
depth Scandinavia is located in a region with an enhanced horizontal temperature gradient, and thus
a strong baroclinicity. Another important region where low-frequency variations of extreme cyclone
depth show significant correlation with other variables is southern Europe. Under high cyclone depth
index conditions, it is a region of minimized meridional temperature gradient, as in the north it is
relatively warm, while the south, i.e. Africa, is characterized by relatively cold temperatures. Such
changes in the temperature field strongly reduce the baroclinic zone, which finally leads to a re-
duction in cyclone frequency over central Europe and to a reduction in cyclone-related precipitation
over southern Europe (Fig. 5c,d, respectively).

Compared to the extreme cyclone depth, which shows distinct and statistically significant corre-
lations, extreme cyclone-related precipitation reveals less clear results. Although an atmospheric
wave train can be identified in the correlation pattern of the 500-hPa geopotential height field, it
shows no statistical significance over the North Atlantic region (therefore not shown). Although
not significant, we find similar patterns in the 1000-hPa geopotential height, indicating that extreme
cyclone-related precipitation may be related to barotropic pressure structures. Furthermore, the 2-m
temperature reveals a slightly significant positive correlation along the European Atlantic coast (Fig.
6a). Thus, enhanced extreme cyclone-related precipitation is related to a warmer coastal line which



leads to increased moisture availability in winter, and thus finally influences the precipitation espe-
cially over Iceland, Scandinavia and the Barents Sea (Fig. 6b).

In summary this analysis shows that different circulation and temperature patterns are related to
extreme cyclone depth and cyclone-related precipitation. Thus, we can conclude that, on average,
cyclones with extreme winds (extreme cyclone depth) are disconnected from cyclones generating

extreme precipitation. Nevertheless, this might not be true for single isolated events.

### 4.4    Anthropogenic forcing impact on cyclone characteristics

Two of the cyclone characteristics (Fig. 4a,e) show strong trends in the 20th and 21st century and thus
are influenced by GHG forcing: the cumulative cyclone presence and the cyclone-related precipita-
tion. In contrast, the wind intensity measured by either central pressure or cyclone depth shows no

significant trend in the 21st century (Fig. 4c). In the following, we discuss these trends with respect
to trends of temperature, precipitation, and cyclone frequency in order to assess potential processes
for GHG induced changes in cyclone characteristics. Further, the relevance of thermodynamic pro-
cesses is investigated by assessing the Clausius-Clapeyron relation.

The temperature trends shown in Fig. 7a are in line with the patterns assessed in IPCC (2013), sug-

gesting a strong warming of the polar areas and the continents and weaker warming of the ocean,
in particular the central North Atlantic shows no significant warming. The former is due to polar
amplification, induced by a strong sea ice reduction and the reduced heat capacity of the land surface
compared to the ocean. The latter is related to changes in the ocean circulation, i.e., a weakening
of the AMOC as projected by most of the comprehensive climate models. These different trends

lead to a change of the horizontal surface temperature gradients, the latter one is a prerequisite for
baroclinicity and thus cyclone development and enhancement. In particular, the contrast between
the North Atlantic and Scandinavia is enhanced, a feature also found in the correlation pattern of ex-
treme cyclone depth with temperature in the period 850 to 1850 (Fig. 5). If similar processes worked
for decadal variations in the Common Era and the future, we would expect to see a positive trend

in extreme cyclone depth, which is not the case in Fig. 4c. Thus, other processes such as increased
static stability (Raible et al., 2010) and the overall decreased meridional temperature gradient – both
reducing cyclones and wind-related intensity – compensate for the locally increased baroclinicity
near Scandinavia.

Precipitation trends also resemble the results presented in the latest IPCC assessment (IPCC, 2013)

showing a negative trend over the Mediterranean and a wetting in high latitudes (Fig. 7b). This is
a first hint that cyclones are redistributed in the future as most of the precipitation in winter in the
mid-latitudes originate from cyclones. Fig. 7c shows the cyclone frequency trend pattern for the 21st
century with significant negative trends mainly over the Mediterranean and partly over the central
North Atlantic. This pattern resembles the precipitation trends and illustrates the connection between



cyclone occurrence and precipitation. As we find a reduction of 12.5 % in the cumulative cyclone
presence over the entire region (Fig. 4a), the reduction over the Mediterranean and the central North
Atlantic cannot be compensated by the positive trends found over Scandinavia and the Hudson Bay
(note that only a small part of the Hudson Bay is included in the area (Fig. 2) of the indices). Again
the signals over the Mediterranean resemble earlier findings obtained with different models (e.g.,

Lionello and Giorgi, 2007; Raible et al., 2010). In these studies, enhanced static stability together
with enhanced stationary wave activity are the main reasons for reduced cyclone activity over the
Mediterranean.

The most striking trend of the cyclone characteristics in Fig. 4 is the positive trend of extreme
cyclone-related precipitation in the 20th and 21st century. The trend pattern of temperature (Fig. 7)

suggests an overall warming, and thus the capability of the air to hold moisture is strongly increased
in the 21st century. To test whether the trend of extreme cyclone-related precipitation is mainly
due to thermodynamics, we estimate the regression coefficients $\beta$ between extreme cyclone-related
precipitation and extreme cyclone-related temperature for the entire simulation in a 150-yr running
window and compare them with the range given by the Clausius-Clapeyron relation, i.e., a 2-3 %

increase in precipitation per 1°C temperature increase (O'Gorman and Schneider, 2009). Note that
similar results are obtained with a 100-yr running window. The regression coefficients for the period
1851-2100 show a strong shift to the upper bound of the Clausius-Clapeyron relation (3 % increase
in precipitation per 1°C), a level never reached during the Common Era (Fig. 8). This is in line
with recent results of Neelin et al. (2017) who found that the interplay of moisture convergence

variance and precipitation loss, increase under global warming. Thus, the result of the 20th and 21st
century agrees with other findings that show that extreme precipitation is mainly thermodynami-
cally driven, as it follows the Clausius-Clapeyron relation under global warming (e.g., Pall et al.,
2007; O'Gorman and Schneider, 2009; Pendergrass and Gerber, 2016; Neelin et al., 2017). Interest-
ingly, we find that roughly 50 % of the periods in the Common Era show a different behavior where

extreme cyclone-related precipitation reacts less to temperature changes than Clausius-Clapeyron
relation would predict, as illustrated by regression coefficients below (0.16 mm/day)/°C. Thus, we
show that the hypothesized general governance of the Clausius-Clapeyron relation on extreme pre-
cipitation (e.g., Pall et al., 2007; O'Gorman and Schneider, 2009) seems to be time dependent. Hints
that this is not just a model result are found in proxy records over Europe, e.g. flood occurrences

cluster also during rather cold periods in the Common Era (e.g., Czymzik et al., 2010; Wirth et al.,
2013; Glur et al., 2013; Amann et al., 2015).

## 5  Conclusions

Extratropical cyclone characteristics are investigated for the period 850 to 2100 CE in a seamless
transient simulation using CESM (version 1) with the focus on the North Atlantic European region



and the winter season (DJF).

The evaluation under present day conditions shows that CESM is able to realistically simulate cyclones and their characteristics, though some biases to the reanalysis product ERA interim remain.

Before 1850, the variability of cyclone characteristics is dominated by internal variability showing pronounced low-frequency variations of different cyclone characteristics. The extreme wind-related
characteristics show a significant connection to the large scale dynamics on decadal time scales, whereas the index representing cyclone-related precipitation is only weakly related to the background temperatures on these time scales. The different cyclone characteristics are not correlated with each other over time, being a first indication that external forcing plays no dominant role in generating these variations. A more detailed analysis of the volcanic and the solar forcing imprint
confirms this and thus earlier findings with other coarsely resolved climate models (Fischer-Bruns et al., 2005).

Future changes are found in two cyclone characteristics. As only one simulation is used in this study it shows that these changes are pronounced and that we should be able to detect these changes at the beginning to mid of the 21th century, irrespective of the realization of natural variability. The
cumulative cyclone presence shows a reduction in the 21st century. This change is already found in studies using coarsely-resolved ensemble simulations with an earlier version of CESM, which compare present day climate with the pre-industrial climate (e.g., Raible et al., 2007). The main decrease of cyclone occurrence is found over the Mediterranean. Using future simulations with another global climate model shows a similar decrease in the Mediterranean (Raible et al., 2010).
The process driving the reduction of cyclones over the Mediterranean is the increase in stability and changes in the stationary wave production over the region in winter (Raible et al., 2010). The other characteristic, which shows a dramatic increase in the future, is the extreme cyclone-related precipitation. This increase is driven by the temperature increase and the Clausius-Clapeyron relation, i.e. purely thermodynamically driven. This is in line with a recent study of Neelin et al. (2017). Thus,
changes in the dynamics seems to be less important for changing precipitation extremes related to winter cyclone activity in the future.

Extending the analysis of the Clausius-Clapeyron relation back in time reveals prolonged periods in the Common Era where extremes do not follow the Clausius-Clapeyron relation. Thus, we hypothesize that in the Common Era both dynamical and thermodynamical processes can be dominant
whereas in the last 100 years and the future under RCP8.5 thermodynamical processes govern extreme events in cyclone-related precipitation. This result is important as many proxy-based studies show that during cold periods of the Common Era hydrological extreme events occur more frequently (Czymzik et al., 2010; Wetter, 2012; Wirth et al., 2013; Glur et al., 2013; Amann et al., 2015). For example, Amann et al. (2015) recently showed in lake sediments that flood occurrences
are enhanced during the LIA, a period known to be cold in Europe – a behavior, which cannot be



explained by the Clausius-Clapeyron relation. As the model simulation in this study also shows periods where the Clausius-Clapeyron relation is unable to explain above-normal extreme cyclone precipitation (14th to 15th century) we hypothesize that these periods were dominated by variability of dynamical processes. Moreover, our simulations show that these variations are mainly driven by
internal variability and that no systematic response to external forcing like during the LIA is evident. So based on our results, the proxies (e.g. Czymzik et al., 2010; Wetter, 2012; Wirth et al., 2013; Glur et al., 2013; Amann et al., 2015) might just show natural internal variability and hence there is no clear justification to interpret them in context of the LIA (i.e., volcanoes and solar forcing).

Thus, future work shall concentrate on processes of low-frequency changes in cyclone character-
istics and the Clausius-Clapeyron relation, e.g., to the assess the role of atmosphere-ocean-sea ice interaction (e.g., Lehner et al., 2013). Further, a more regional view on Europe is needed to focus on impacts on land relevant for insurance providers.

*Acknowledgements.* This work is supported by the Swiss National Science Foundation (grant: 18-001). The CESM simulation is performed on the super computing architecture of the Swiss National Supercomputing
Centre (CSCS).





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





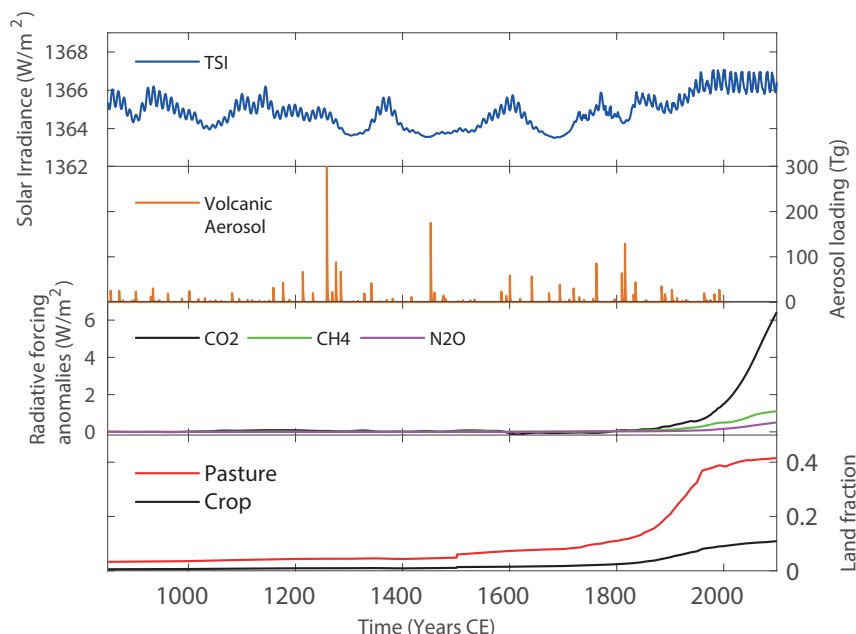

**Fig. 1.** Forcings used in the last millennium simulation with CESM. From Top to bottom: total solar irradiance (TSI), total volcanic aerosol mass; radiative forcing (calculated according to IPCC, 2001) from the greenhouse gases $CO_2$, $CH_4$, and $N_2O$; and major changes in land cover as fraction of global land area. The figure is adapted from Lehner et al. (2015).

**Table 1.** Correlation between different cyclone characteristics. The upper right of the table represent CESM correlation, the lower left ERA interim. Bold numbers indicate significant correlation at the 5 % level using a two-side student $T$ test. For this analysis the central SLP time series is multiplied by $-1$ so that low central pressure corresponds to a high cyclone depth resulting in a positive correlation.

|  | Cyclone time steps | Radius | Cyclone depth | SLP | Cyclone rel. Precipitation |
|---|---|---|---|---|---|
| cyclone time steps | 1 | 0.21 | **-0.42** | **-0.49** | 0.11 |
| Radius | **0.42** | 1 | **0.47** | -0.36 | 0.07 |
| Cyclone depth | **-0.4** | 0.09 | 1 | **0.63** | 0.16 |
| SLP | **-0.58** | 0.07 | **0.81** | 1 | 0.13 |
| Cyclone rel. Precipitation | -0.27 | -0.11 | 0.22 | 0.15 | 1 |





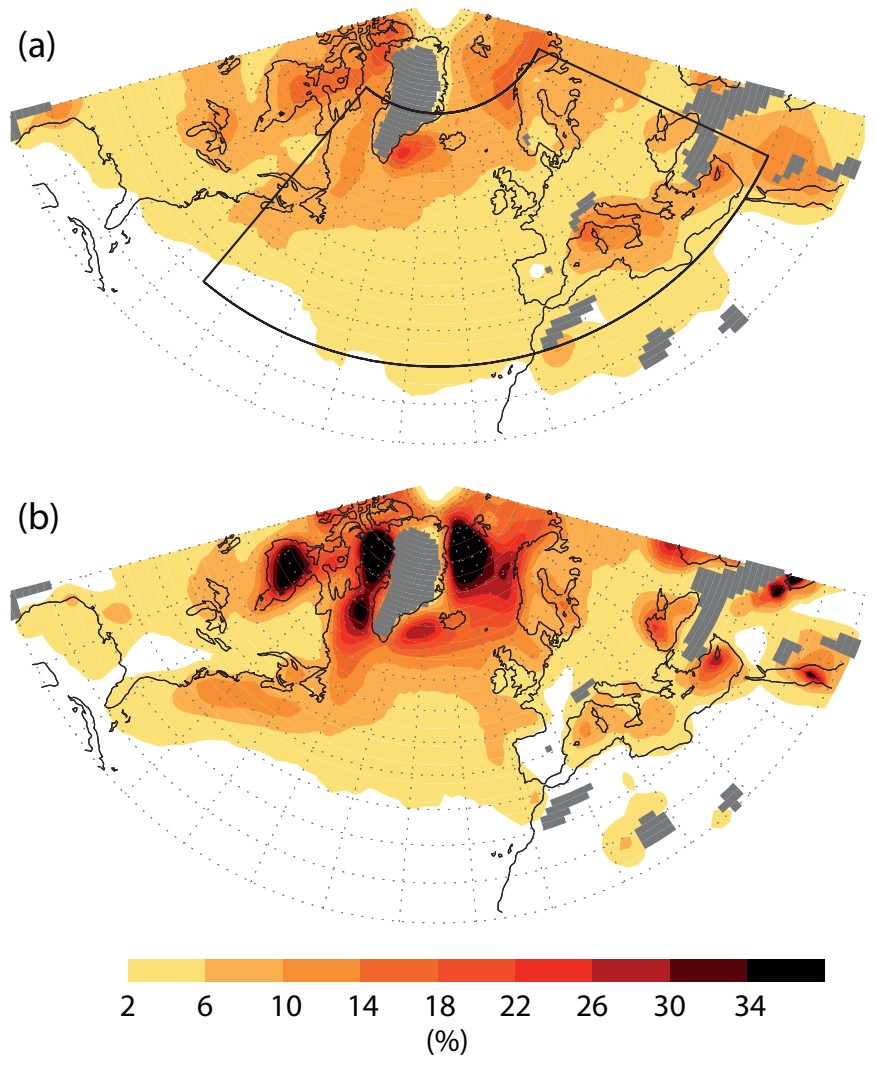

**Fig. 2.** Cyclone center frequency (% presence per season) for (a) the ERA interim and (b) the CESM simulation for the period AD 1980-2009 in winter (DJF). The bounded domain illustrates the region North Atlantic used to estimate different cyclone characteristics. Grey areas are higher than 1 km above sea level and are excluded from the cyclone detection and tracking method. 10% presence per season means that in 10% of the winter season a cyclone is present at a grid point.

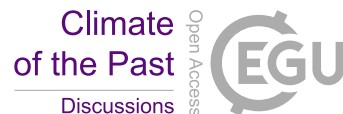



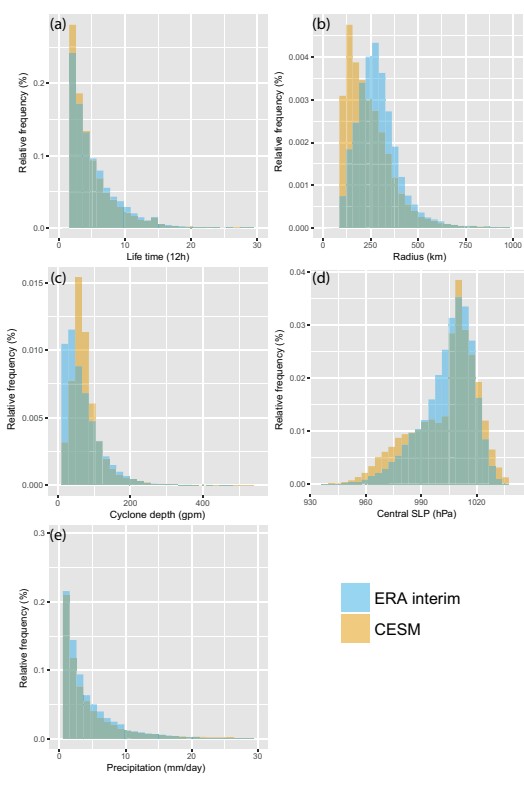

**Fig. 3.** Histograms of different cyclone characteristics: (a) life time of cyclones, (b) radius of cyclones, and (c) cyclone depth, (d) central SLP and (e) cyclone-related precipitation. The histograms are based on the cyclones detected from 1981-2010, i.e., 7624 cyclones in ERA interim and 12369 in CESM.




**Fig. 4.** Long-term time behavior of different cyclone characteristics illustrated by time series averaged with 30-yr running window: (a) cumulative cyclone presence, (b) median radius of the cyclones, and 90th percentile of (c) cyclone depth, (d) central SLP and (e) cyclone-related precipitation.





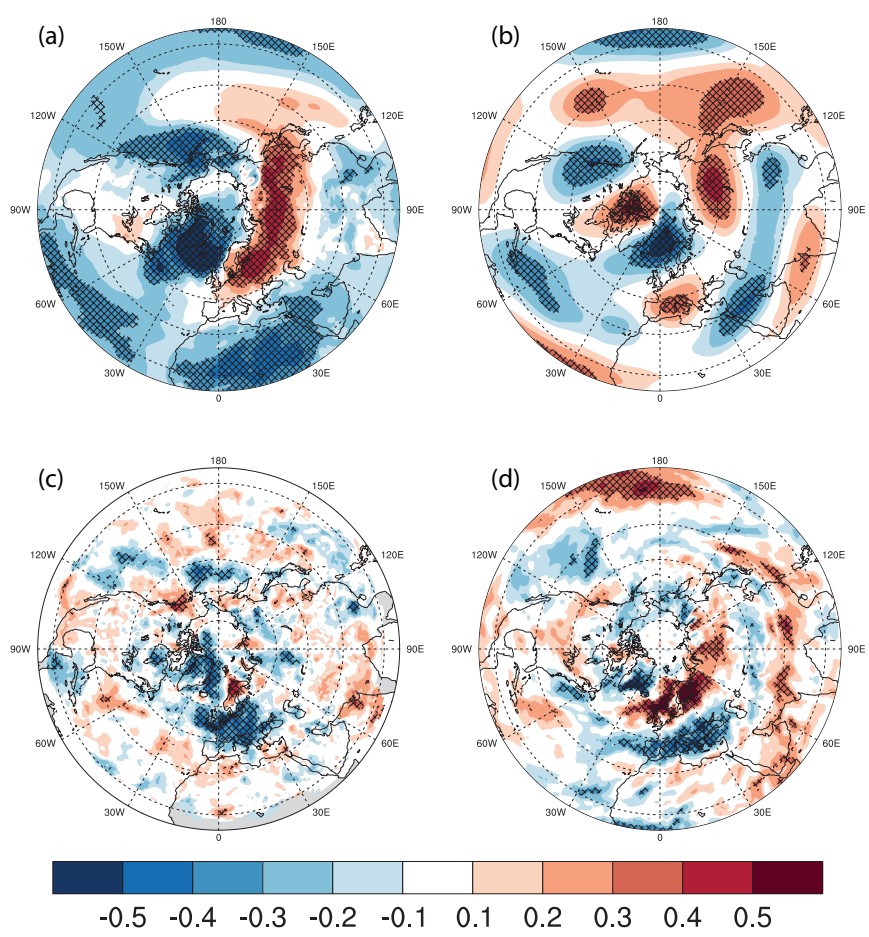

**Fig. 5.** Correlation between the 30-yr running mean times series of extreme cyclone depth and (a) 2-m temperature, (b) 500-hPa geopotential height, (c) cyclone frequency, and (d) precipitation for the period 850-1850 CE. The 5 % significance level using a student $T$-test is illustrated with cross-hatching.





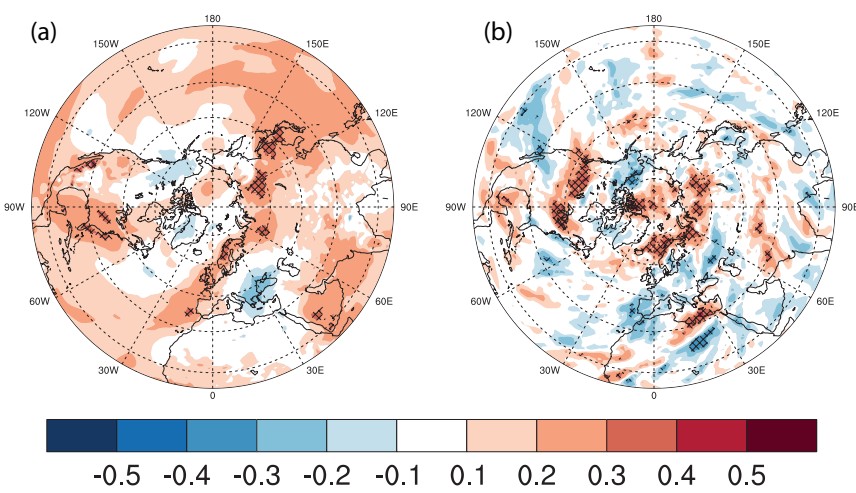

**Fig. 6.** Correlation between the 30-yr running mean times series of extreme cyclone-related precipitation and
(a) 2-m temperature and (b) precipitation for the period 850-1850 CE. The 5 % significance level using a student
$T$-test is illustrated with cross-hatching.



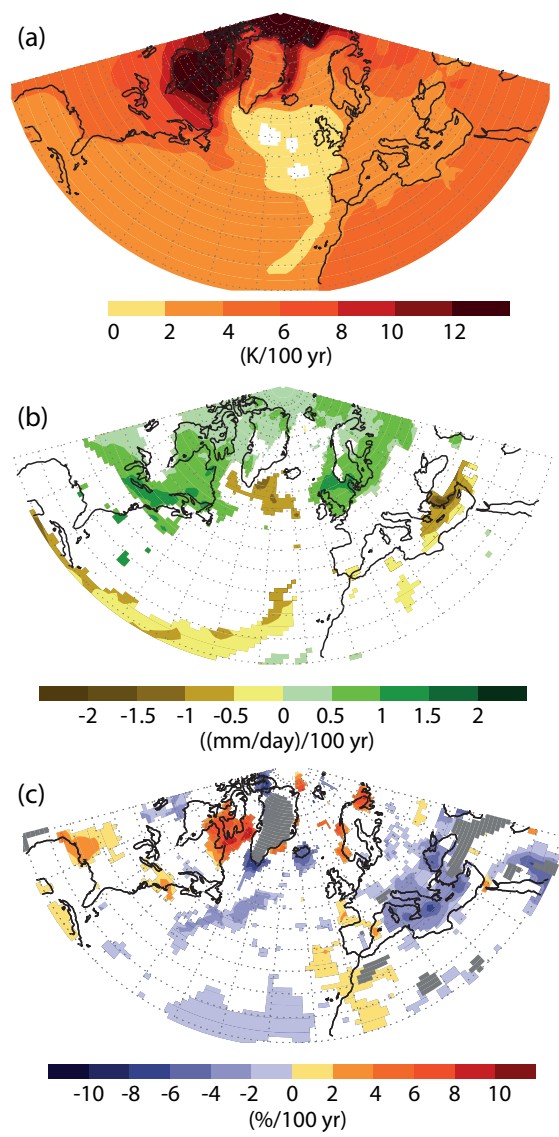

**Fig. 7.** Trends from 2005 to 2100 for (a) 2-m temperature, (b) precipitation, and (c) cyclone frequency. Only significant trends at the 5 % significance level using a student $T$-test are shaded. Grey areas in (c) are excluded from the cyclone detection and tracking method (1000 m a.s.l.).





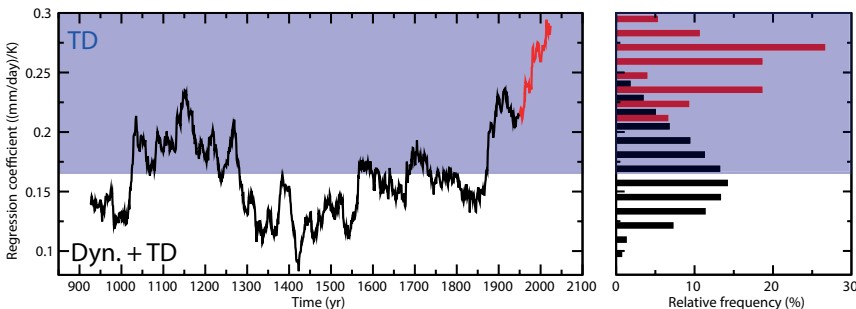

**Fig. 8.** Time series of regression coefficient estimated between mean cyclone-related temperature and 90th percentile of cyclone-related precipitation for 150-yr window running through the entire period 850 to 2100: Common Era (black) and era influenced by RPC8.5 (red). The right panel shows the histogram of the regression coefficients for the two periods (bin width 0.01). All periods within the blue shading follow the Clausius-Clapeyron relation, named thermodynamic (TD) (O'Gorman and Schneider, 2009). In the white area, dynamics (Dyn.) and thermodynamics are relevant.