# Peer review of "Extratropical cyclone statistics during the last millennium and the 21st century"

_Climate of the Past, 2018_

## Referee Comment (RC1) · Anonymous Referee #1 · 4 Jul 2018

This paper documented the variability of Extratropical Cyclones (EC) in a 1-deg CESM model during the last millennium and the 21st century under RCP8.5 forcing. They found natural variations on decadal and multi-decadal timescales and substantial changes in cyclone frequency and precipitation under anthropogenic forcing. It is also found that the cyclone-related precipitation changes do not always conform to the C-C relationship. The paper is presented in a clear and organized manner. Although most of the analyses are quite generalized and most of the results should be expected, studies of EC in such long-term historical simulations aren't common. I think this could serve as an introductory paper for more detailed studies using the same or similar simulations in the future.

My main comment is about the regression analysis applied in Section 4.3. It seems

that the regression is based on the average depth/precipitation index of the entire North Atlantic. Because of the spatial average, it is no surprise that only the most dominant and large-scale circulation pattern (i.e., the NAO) would show up in Figure 5. While such analyses are convenient and easily make sense, the mechanisms that cause the spatial variations of ECs are muddled. Considering the large spatial variability of EC, I would prefer to see results that do not just focus on the regional mean. For example, the authors could try applying EOF analyses to the data to extract some spatial information.

Some minor comments:

Line 36: "millennium which" -> "millennium, which", and similar changes throughout

Line 37-38: "the external forcing . . . characteristics" – awkward sentence. Suggest changing it to "the externally forced and internally varying extratropical cyclone activities"

Line 135: Would the cyclone statistics change if higher frequency outputs are used? What if time averaged instead of instantaneous outputs are used? Is the ERA data also 12-h instantaneous?

Line 201: "area North" -> "area of North"

Line 221: "as well as with" -> "as well as between"

Line 267: any references for the improvements?

Line 282: "missing volcanic forcing impact". Do all the volcanic forcings have the same spatial patterns? It is likely that the effect of volcanoes cancels out due to their varying spatial patterns, but the individual volcanoes may still be impactful.

Line 322: "region of minimized" -> "region of reduced"

Line 355: "change of the" -> "change in"; "the latter one" -> "which"

---

## Referee Comment (RC2) · Anonymous Referee #2 · 7 Jul 2018

Formal review of manuscript for Climate of the Past

Manuscript identification number: CP-2018-58

Title: Extratropical cyclone statistics during the last millennium and the 21st century

Authors: C.C. Raible, M. Messmer, F. Lehner, T.F. Stocker, R. Blender

Recommendation: Minor revision

General Comments: The authors investigate the variability of extra-tropical cyclone characteristics for the North Atlantic / European region based on a long coupled GCM simulation (850-2100). First, the variability pre-1850 is evaluated, rendering the general result that in spite of the identified multi-decadal variability no external forcing imprint

is identified for this period. On the other hand, a general decrease in cyclone numbers (particularly for the Mediterranean) and cyclone related precipitation (e.g. north of 50°N over Europe) is identified for the XXI century. Finally, the authors discuss the possible relevance of thermodynamic vs dynamical processes for the identified trends / variability. The manuscript is well written, the methodologies and statistics are well applied, and the conclusions are largely sound. The consideration of such a long transient run is quite unusual, and the embedded discussion of natural vs anthropogenic forcing is quite interesting. Therefore, I believe the manuscript is a worthy contribution to Climates of the Past. Nevertheless, several minor aspects should be improved / better discussed before the paper is in acceptable form. Therefore, I recommend a minor revision according to the comments given below.

Minor Comments:

**1: lines 47-58: There is quite a lot of additional literature in this topic, so I understand the authors need to do a selection. However, I would recommend to include the two review papers of Ulbrich et al. (2009) and Feser et al. (2015), e.g. on line 48 and 52. If possible, a few more sentences on the different measures of cyclone activity and the regional differences would be excellent.**

**2: line 60: Please clearly state here that you mean that the low level meridional temperature gradients are reduced on average. On the upper troposphere, it is the opposite, as the strongest warming occurs in the tropical regions. Please find a suggestion below. This should also be stated more clearly other text passages.**

"The decrease of the projected low level meridional temperature gradient on average (due to strong high latitude near surface warming associated with polar amplification) implies a decrease of storm activity in the future, (. . .)<"

**3: lines 161-163: While I understand the authors′ idea to consider the 90th percentile of central pressure and cyclone depth as a proxy for windiness, I think it would have been easy to assign peak near surface wind speeds close to the cyclone core (e.g.**

Zappa et al., 2013) a more adequate measure of windiness associated with the cyclones. What has this not been done? Was the near-surface wind data not stored? Or was there another reason? This potential shortcoming makes a few statements in the manuscript (e.g. line 338-340) less robust and should at least be discussed as a potential shortcoming.

**4: lines 185-199: It is a bit unusual that the (lower resolution) GCM has a higher cyclone frequency as the ERA-Interim dataset. While I tend to agree with the authors that this may be partially associated with an enhanced number of weak lows in the GCM, I wonder in how far the (bi-linear!!!) re-gridding of the ERA-Interim played a role here. What do the cyclone statistics with the original ERA-interim grid look like? Are the statistics more comparable if one only considers strong cyclones (e.g. exceeding a certain depth)?**

**5: lines 301-310: Given that the main author has co-written a review paper on the NAO variability during the last millennium (Pinto and Raible, 2012), I wonder why so little is discussed about the link well established link between the NAO variability and cyclone variability over the Eastern North Atlantic and Europe (except for this text passage). In my opinion it would be pertinent to strengthen this statement and discuss a bit in how far the NAO variability in the simulation matches (or not) the cyclone variability for various parameters shown in Fig. 4, and in how far this agrees with NAO reconstructions. Even if the authors will surely explore this further in subsequent (and more regional) studies in the future, I suggest expanding the topic a bit here.**

**6: lines 378-379: I suggest referring to Zappa et al (2013) here, which showed exactly this based on the CMIP5 model ensemble.**

**7: lines 381-401: The interesting thing here is that the increase in cyclone related precipitation is particularly clear north of 50°N (notably over Europe), while elsewhere reduced precipitation is often found, particularly at lower latitudes. Recent studies (e.g. Santos et al. 2016) have identified that there may be a "circulation independent" increase of precipitation north of $\sim 45°$N over Western Europe and comparative drying around 35-45°N (cf. their Figure 9). This may imply that for the latter the increase of humidity is overcompensated by temperature (thus lower relative humidity) or hampered by increased subsidence. I think that present statement for the whole region regarding the Clausius-Clapeyron relationship is too general , and a more differentiated regional discussion would be quite interesting.**

**11: line 424: Please add Ulbrich et al (2009) and Zappa et al. (2013) here.**

**12: line 429: see discussion in #10, please enhance, maybe adding "at least north of 50°N" or similar.**

References (not exhaustive):

Feser F., et al. (2015). Storminess over the North Atlantic and Northwestern Europe: A review. QJRMS, 141, 350-382. doi:10.1002/qj.2364.

Pinto JG, Raible CC (2012) Past and recent changes in the North Atlantic Oscillation. WCC, 3, 79–90. doi:10.1002/wcc.150

Santos JA, (2016) Understanding climate change projections for precipitation over Western Europe with a weather typing approach. JGR-A, 121, 1170–1189. doi:10.1002/2015JD024399

Ulbrich U, et al (2009) Extra-tropical cyclones in the present and future climate: a review. TAAC, 96, 117–131. doi:10.1007/s00704-008-0083-8

Zappa G., et al. (2013) A multimodel assessment of future projections of North Atlantic and European extratropical cyclones in the CMIP5 climate models. JCLIM, 26, 5846–5862. doi:10.1175/JCLI-D-12-00573.1, 2013.

---

## Referee Comment (RC3) · Anonymous Referee #3 · 11 Jul 2018

This manuscript investigates a long integration of the 1deg version of the CESM, from 850-2100 (with RCP8.5 forcings). The authors use 12-hourly data to track extratropical cyclones (ETCs) over the North Atlantic. Results are dominated by interannual-to-decadal scale fluctuations of cumulative ETC metrics (count, intensity, precipitation) but no obvious external forcing signal is noted. After 2100, strong increases in ETC precipitation and decreases in ETC count are noted, with authors applying a regression analysis to demonstrate that these changes are mostly thermodynamic in nature, in line with previously published work. Some regional variations are also considered, particularly over the Mediterranean and Scandinavia.

In general, I feel the manuscript is clear and crisp, albeit not with overly novel conclusions. As a scientist who deals mostly with future storminess associated with climate

change, I think this type of analysis is relevant to our understanding of climate models and the dynamics of the features themselves within the climate system over long time periods. Where I do have one concern is the results of the tracking algorithm, particularly with regards to CESM, that may be somewhat influencing the results. Before final publication, I feel these should be addressed by either retracking the storms or running a sensitivity analysis. Assuming the authors have a pipeline that performs the subsequent analysis in Figs. 4-8, this should be fairly trivial to undertake.

As one who has used CESM data in the past, if the authors are using the in-line 1000hPa geopotential (Z1000) as a variable (versus calculating Z1000 using the hybrid coefficients and topography) they are likely having issues with the fact that CESM will not automatically interpolate "below ground." Therefore, while the true Z1000 is likely negative over high-terrain areas (e.g., Greenland) the Z1000 reported from CESM is anomalously positive since the code will not go below the lowest model level (at least, according to my recollection). This is a quirk of the CESM in-line interpolation and is likely causing the issues (high cyclone count near high-terrain areas) seen in Fig. 2b since the "background" Z1000 field is biased very high. This can probably be rapidly verified by just comparing the time-mean Z1000 in both ERA-Interim (ERA-I) and CESM. The optimal correction for this would be to use some sort of offline solver with the 3-D Z field and Python/NCL/IDL/etc.

In this vein, it is not clear why the authors are not tracking on sea level pressure (PSL), which is essentially a prognostic quantity in most climate models (technically PS is prognostic, but the correction to PSL primarily uses other prognostic variables like T and the surface topography field). PSL is a much more widely-used quantity when evaluating climate models and would likely alleviate the issues

The fact that CESM simulates far more cyclones than ERA-I is therefore questionable. While there are certainly some differences in effective resolution, etc. of the datasets, a factor of almost 2x (Line 202) in the total number of storms between CESM and ERA-I seems quite high at first blush. The authors hypothesize this is due to weak

storms, but that is not clear to me from Fig. 3. For example, the SLP distribution shows more weak storms for CESM, but also more strong ones. Having a smaller radius distribution is also not necessarily indicative of weaker storms, as aspects of the model configuration such as numerical diffusion and how grids are interpolated may contribute to differences here. This is somewhat hinted at in Figs 3c-d. As an additional example, one could make an argument that 1deg ETCs would be "smaller" than 4deg ETCs, but 1deg ETCs *should* be more intense based on being better resolved.

I would like the authors to consider "retracking" the storms if PSL is available. They could easily modify their algorithm to search for prognostic deficits in PSL as in other trackers within the IMILAST project (of which the lead authors of this manuscript already contributed to). If that is not available, I would like the authors to try and evaluate whether or not the issues of additional ETCs tracked in CESM are related to the Z1000 issue noted above. One option would be to run CESM for a short period (perhaps a few decades) and compare the results of using the inline Z1000 with PSL or a more accurately diagnosed Z1000.

Minor comments:

Line 122: "So called" is too colloquial, would just say "this is the 1deg version of the model used in CMIP-class experiments" or thereabouts.

Line 123-124: Would include a sentence or two about the subgrid physics package used in this version of CESM (in the atmospheric model) since that would have the largest impact on the results here, particularly thermodynamic ones.

Line 245: Are there changes in mean storm-track, basin-wide surface pressure, etc. that may be relevant here?

Line 262: 4deg models certainly underresolve synoptic scale features, which ETCs are.

Line 299: Is this a basin-wide metric? I question a bit about correlating the spatial

pattern with basin-wide metrics as then I'd expect large scale North Atlantic patterns that control ETCs to dominate this analysis (e.g., the NAO).

Line 312: The spatial field remains quite noisy, I would be a bit careful about being too conclusive since, even with a multi-century simulation, I'm not sure we can be completely confident very small (O(10deg)) spatial patterns are tremendously significant in a model whose effective resolution is probably ∼6deg (see Skamarock 2004 for discussion of effective resolution in numerical models).

Line 321: This reads as a bit "hand-wavy;" I'd formalize and clean up the text a bit.

Line 332: These "barotropic pressure structures" could be underresolved warm core storms (e.g., tropical cyclones) moving to mid-to-high latitudes. 1deg models are capable of starting to simulate these features, albeit far weaker than what is observed (e.g., Wehner et al., 2014, Walsh et al., 2015).

References:

Skamarock, W.C., 2004: Evaluating Mesoscale NWP Models Using Kinetic Energy Spectra. Mon. Wea. Rev., 132, 3019–3032, https://doi.org/10.1175/MWR2830.1

Walsh, K.J., S.J. Camargo, G.A. Vecchi, A.S. Daloz, J. Elsner, K. Emanuel, M. Horn, Y. Lim, M. Roberts, C. Patricola, E. Scoccimarro, A.H. Sobel, S. Strazzo, G. Villarini, M. Wehner, M. Zhao, J.P. Kossin, T. LaRow, K. Oouchi, S. Schubert, H. Wang, J. Bacmeister, P. Chang, F. Chauvin, C. Jablonowski, A. Kumar, H. Murakami, T. Ose, K.A. Reed, R. Saravanan, Y. Yamada, C.M. Zarzycki, P.L. Vidale, J.A. Jonas, and N. Henderson, 2015: Hurricanes and Climate: The U.S. CLIVAR Working Group on Hurricanes. Bull. Amer. Meteor. Soc., 96, 997–1017, https://doi.org/10.1175/BAMS-D-13-00242.1

Wehner, M. F., K. A. Reed, F. Li, Prabhat, J. Bacmeister, C. Chen, C. Paciorek, P. J. Gleckler, K. R. Sperber, W. D. Collins, A. Gettelman, and C. Jablonowski (2014), The effect of horizontal resolution on simulation quality in the Community Atmospheric Model, CAM5.1, J. Adv. Model. Earth Syst., 6, 980–997, doi: 10.1002/2013MS000276.

---

## Referee Comment (RC4) · Anonymous Referee #4 · 13 Jul 2018

I have reviewed this manuscript cp-2018-58 entitled "Extratropical cyclone statistics during the last millennium and the 21st century" by Christoph C. Raible et al. In this manuscript, the authors studied the extratropical cyclones and their changes in the 20th and 21st centuries using a unique CESM1 simulation from 850-2100 with high temporal output. They found that the variations of the cyclones over the North Atlantic and Europe sector before industrialization are mainly related to the internal variability, not directly related to either volcanic or solar forcing (nature forcing). Towards the 21st century, two of the cyclone metrics show significant trends. They also show that the Clausius-Clapeyron relation is not always followed by the changes of the cyclones and the global mean temperature. I found this manuscript is very interesting and worth to be published subject to some minor revision.

Comments:

1. the authors using a 30-year running correlation to show whether changes of solar forcing will affect the cyclone activity and found there is no relationship. Since one of the major solar cycle is 11-years, with a 30-year running window, it will not show the effect of solar forcing on the cyclone activities. The authors could do a spectrum analysis for the cyclone activities and check with the solar forcing cycles (11-year or other cycles). This could give a better sense on whether solar activity would or would not affect the cyclones. 2. Line 351, "The former is due to polar amplification, induced by a strong sea ice reduction and the reduced heat capacity of the land surface compared to the ocean", in this sentence, it is not clear to me why the heat capacity of land reduces? it is because the ocean heat capacity increases? or something else. It would be nice that the authors could explain this a bit better or add some references. 3. Overall, it seems that the physical explanations are a bit weak in this manuscript.

―――――――――――――――

---

## Author Comment (AC1) · 27 Jul 2018

Dear Reviewer

Please find below our reply to your review in the discussion process of our manuscript. Please note that this reply is not the comprehensive point-to-point response which will be due in a later stage of the review process (depending on the editor's decision). Thus, we concentrate on the major comments and try to show how we will deal with the helpful suggestions and questions raised.

The reviewer listed some minor comments which we will handle with care in the revised version of the manuscript. Clearly, we will add an analysis of the 11 yr cycle, though our expectation is that we will not see a signal as in other analysis of the mean circulation

signals of the 11 yr cycle were not detected.

Furthermore, we will try to be more precise in the revised version of the manuscript and add more physical explanations of the processes involved.

Thank you again for the helpful comments which certainly will improve our manuscript.

Sincerely yours,

Christoph Raible

on behalf of the author team

---

## Author Comment (AC2) · 27 Jul 2018

Dear Reviewer

Please find below our reply to your review in the discussion process of our manuscript. Please note that this reply is not the comprehensive point-to-point response which will be due in a later stage of the review process (depending on the editor's decision). Thus, we concentrate on the major comments and try to show how we will deal with the helpful suggestions and questions raised.

'The reviewer suggests to redo the analysis by using SLP instead of Z1000. The main reason is that the reviewer thinks we have used the in-line 1000hPa geopotential (Z1000) as a variable (versus calculating Z1000 using the hybrid coefficients

and topography). Actually we have calculated Z1000 offline from the 3-dimensional geopotential height field (See ncl script below). Comparing the Z1000 with the SLP averaged for the period 1980-2009 we still see a small positive difference over the center of Greenland (Fig 1, reply, below). However, this difference between Z1000 and SLP does not affect the two regions where CESM overestimates cyclones (see manuscript Fig. 2), namely in the Hudson Bay and in between Iceland and Spitzbergen. We additionally made a visual test (movie of Z1000 field) and see that CESM simulates more cyclones traveling to these two regions than ERA interim. So there are no stationary cyclones in these regions which may be a hint that the interpolation of Z1000 leads to some artificial cyclones. Note also the effect of the slightly coarser resolution of ERA interim (here 1.5 degree) which lead to a less pronounced meridional pressure gradient in the North Atlantic. So some of the bias might be due to weak cyclones (we will give more arguments for this in the revised version), but certainly CESM also overestimates cyclones. We will be more clear on this fact in the revised version.

Furthermore, we will take care on the minor comments and add the literature, in particular concerning the subgrid physics and give some reasoning on the use of basin wide storm track measure and how the can be interpreted. The main reason is that we focus here on the broader picture and certainly plan to perform a regional focus on regions in Europe in a next publication.

Thank you again for the helpful comments which certainly will improve our manuscript.

Sincerely yours,

Christoph Raible

on behalf of the author team

NCL script:

.*********************************************
'

; This script interpolates sigma to pressure coordinates
```
; in outputs the geopotential height (for all plevs chosen)
;***********************************************
;
load "$NCARG_ROOT/lib/ncarg/nclscripts/csm/gsn_code.ncl"
load "$NCARG_ROOT/lib/ncarg/nclscripts/csm/gsn_csm.ncl"
;***********************************************
;
begin
;*************************************************
; read in data
;*************************************************
;
;******only years
;******year has to be given with the function call: 'for a in { }; do ncl
;******year=... Interpolation_... ;done'
path1="/BPRD_trans/atm/hist/BPRD_trans.cam2.h1.$yy-01-01-00000.4.nc"
in = addfile(path1,"r")
;————————————————————————————————————————————-
; read needed variables from file
;————————————————————————————————————————————-
Z3 = in->Z3 ; select variable to be converted
P0mb = 1000.
hyam = in->hyam ; get a coefficiants
hybm = in->hybm ; get b coefficients
```

```
PS = in->PS ; get pressure

TBOT = in->TS ; get temperature at lowest layer (closest to surface)

dims = dimsizes(Z3)

nlevs= dims(1)

PHIS = Z3(:,nlevs-1,:,:)*9.81 ; get geopotential [m^2/s^2] at the bottom (lowest layer)

;————————————————————————————————————-

; define other arguments required by vinth2p

;————————————————————————————————————-

; type of interpolation: 1 = linear, 2 = log, 3 = loglog

interp = 2

; is extrapolation desired if data is outside the range of PS

; extrap = False

extrap = True

; A scalar integer indicating which variable to interpolate: 1 = temperature,

; -1 = geopotential height, 0 = all others.

varflg = -1

; create an array of desired pressure levels:

plevs =(/ 1000.0 /)

plevs!0 = "plevs"

plevs&plevs = plevs

plevs@long_name = "Pressure"
```

```
plevs@unit = "hPa"
intVar_PS = vinth2p_ecmwf(Z3,hyam,hybm,plevs,PS,interp,P0mb,1,extrap,varflg,TBOT,PHIS)
intVar_PS!0 = "time"
intVar_PS!1 = "lev"
intVar_PS!2 = "lat"
intVar_PS!3 = "lon"
intVar_PS&time = in->time
intVar_PS&lev = plevs
intVar_PS&lat = in->lat
intVar_PS&lon = in->lon
intVar_PS@units = "m"
intVar_PS@long_name = "Geopotential Height (above sea level)"
;system("echo saving")
;setfileoption("nc","Format","NetCDF4Classic")
;fileout=getenv("FZ")
fileout="BPRD_trans.cam2.h1.$yy-01-01.z1000.nc"
system("rm " + fileout) ; remove any pre-existing file
fout=addfile(fileout,"c")
fout->Z3 = intVar_PS ; write into new file
system("echo new file for GPH") ; print path and new file to screen as confirmation
```

[Figure]

[Figure]

**Fig. 1.** Comparison of mean SLP and Z1000 field for winter DJF for the period 1979 to 2009: (top) CESM, (bottom) ERA interim.

---

## Author Comment (AC3) · 27 Jul 2018

Dear Reviewer

Please find below our reply to your review in the discussion process of our manuscript. Please note that this reply is not the comprehensive point-to-point response which will be due in a later stage of the review process (depending on the editor's decision). Thus, we concentrate on the major comments and try to show how we will deal with the helpful suggestions and questions raised.

We will include the minor comments raised by the reviewer. In particular, we will enlarge the literature review, make clear that we focus in the discussion on temperature gradients at lower levels. We will also add a discussion on the bi-linear re-gridding of

[Figure]

ERA interim, and the NAO and will enlarge the discussion of the CC relationship to regional scales. Additionally, we will test the statistics for a fixed cyclone depth threshold. Unfortunately, we cannot use 10-m wind directly as the data has not been stored. Note that 10-m wind is a diagnostic variable which may also suffer some shortcomings. We will include some discussion on potential shortcomings but given the fact that the model has a resolution of 1 degree, important mesoscale phenomenon like the sting jet will be anyway not resolved.

Thank you again for the helpful comments which certainly will improve our manuscript.

Sincerely yours,

Christoph Raible

on behalf of the author team

---

## Author Comment (AC4) · 31 Jul 2018

Dear Reviewer

Please find below our reply to your review in the discussion process of our manuscript. Please note that this reply is not the comprehensive point-to-point response which will be due in a later stage of the review process (depending on the editor's decision). Thus, we concentrate on the major comments and try to show how we will deal with the helpful suggestions and questions raised.

The main comment concerns the regression analysis applied in Section 4.3. Our aim with this analysis was to give a broad picture of how mean (over the North Atlantic) cyclone characteristics are connected to other variables illustrating e.g. the mean circulation. The reviewer suggests to apply EOF analysis to the different cyclone characteristics and then assess the spatial pattern obtained in more details. This is in principle a very interesting idea and we thought about it when performing the analysis. Still there are some issues of concern. The cyclone characteristics make not much sense on the grid point scale (they are mostly measures related to the cyclone center and there are certainly grid points where no cyclones have traveled over), so some area averaging is necessary (e.g. using all grid points in a circle of 500 or 1000 km around a point). This averaging would induce some spatial dependency which will affect the EOF analysis. For the moment we thought that such an analysis is a story of its own and is a bit beyond the scope of study to give a broad picture. Certainly, we have the plan to focus more on the impacts of Europe in an accompanied study and separate Europe in northern, southern and central part. Still to improve the current manuscript we will try to discuss in more details which mechanism can be obtain from such an analysis and which not.

Furthermore, the reviewer added several very helpful minor comments. We will include the minor comments in the revised version of the manuscript and clarify the raised questions on the model output and the volcanic imprint.

Thank you again for the helpful comments which certainly will improve our manuscript.

Sincerely yours,

Christoph Raible

on behalf of the author team

---

## Author Response (AR1)

**Reply to Extratropicalcyclones by Raible et al**

**Anonymous Referee #1**

This paper documented the variability of Extratropical Cyclones (EC) in a 1-deg CESM model during the last millennium and the 21st century under RCP8.5 forcing. They found natural variations on decadal and multi-decadal timescales and substantial changes in cyclone frequency and precipitation under anthropogenic forcing. It is also found that the cyclone-related precipitation changes do not always conform to the CC relationship. The paper is presented in a clear and organized manner. Although most of the analyses are quite generalized and most of the results should be expected, studies of EC in such long-term historical simulations aren't common. I think this could serve as an introductory paper for more detailed studies using the same or similar simulations in the future.

My main comment is about the regression analysis applied in Section 4.3. It seems that the regression is based on the average depth/precipitation index of the entire North Atlantic. Because of the spatial average, it is no surprise that only the most dominant and large-scale circulation pattern (i.e., the NAO) would show up in Figure 5. While such analyses are convenient and easily make sense, the mechanisms that cause the spatial variations of ECs are muddled. Considering the large spatial variability of EC, I would prefer to see results that do not just focus on the regional mean. For example, the authors could try applying EOF analyses to the data to extract some spatial information.

The aim of the regression analysis was to give a broad picture of how mean (over the North Atlantic) cyclone characteristics are connected to other variables illustrating e.g. the mean circulation. The reviewer suggests to apply EOF analysis to the different cyclone characteristics and then assess the spatial pattern obtained in more details. This is in principle a very interesting idea and we thought about it when performing the analysis. Still, there are some issues of concern. The cyclone characteristics make not much sense on the grid point scale (they are mostly measures related to the cyclone center and there are certainly grid points where no cyclones have traveled over which makes the fields very noisy), so some area averaging is necessary (e.g. using all grid points in a circle of 500 or 1000 km around a point). This averaging would induce some spatial dependency which will affect the EOF analysis. Another way to treat this problem is to focus on specific 'impact regions', e.g., Central, Northern or Southern Europe and assess the different cyclone characteristics for these regions and perform a regression-correlation or composite analysis. For the moment we think that such an analysis is a story of its own and is beyond the scope of this study (i.e., to give a broader picture overview). Certainly, we have the plan to focus more on the impacts over Europe in an accompanied study as mentioned at the end of the manuscript. Still, to improve the current manuscript we discuss in more details which mechanistic understanding can be obtained from such an analysis and which not. This is done at the end of section 4.3 and in the conclusions.

Some minor comments:

Line 36: "millennium which" -> "millennium, which", and similar changes throughout

Applied throughout the manuscript as suggested.

Line 37-38: "the external forcing … characteristics" – awkward sentence. Suggest changing it to "the externally forced and internally varying extratropical cyclone activities"

Done as suggested.

Line 135: Would the cyclone statistics change if higher frequency outputs are used? What if time averaged instead of instantaneous outputs are used? Is the ERA data also 12-h instantaneous?

There are several studies which investigated the influence of the higher frequency output. Clearly, the cyclone can be identified more precisely and more cyclones are found when one uses higher frequency output or higher spatially resolved data, as suggested by Blender and Schubert (2000). Though we decided to save only 12-h output for reasons of data storage. It is not common to use time averaged output as this would blur the cyclone center.
We clarified the selection of 12-h output and also added that we have used 12-h instantaneous output of ERA interim.

Line 201: "area North" -> "area of North"

Changed to 'We focus on the North Atlantic…'

Line 221: "as well as with" -> "as well as between"

Done

Line 267: any references for the improvements?

We relate the statement mainly to the higher resolution, but also several relevant processes in CAM4 (atmospheric part of CESM1) were improved, so we added the reference Neale et al. (2013).

Line 282: "missing volcanic forcing impact". Do all the volcanic forcings have the same spatial patterns? It is likely that the effect of volcanoes cancels out due to their varying spatial patterns, but the individual volcanoes may still be impactful.

The spatial patterns of the volcanic eruptions are different. As we focus on the strongest eruptions they are all of tropical origin, which diminish the effect mentioned. We also individually analyzed the strongest eruptions and did not find a clear signal in the cyclone characteristics. Note that we focused our analysis on the extremes.
We clarified that the superposed epoch analysis focus on tropical eruptions.

Line 322: "region of minimized" -> "region of reduced"
Done

Line 355: "change of the" -> "change in"; "the latter one" -> "which"

Done

Thank you again for the helpful comments.

**Anonymous Referee #2**

Formal review of manuscript for Climate of the Past
Manuscript identification number: CP-2018-58
Title: Extratropical cyclone statistics during the last millennium and the 21st century
Authors: C.C. Raible, M. Messmer, F. Lehner, T.F. Stocker, R. Blender
Recommendation: Minor revision

General Comments:

The authors investigate the variability of extra-tropical cyclone characteristics for the North Atlantic / European region based on a long coupled GCM simulation (850-2100). First, the variability pre-1850 is evaluated, rendering the general result that in spite of the identified multi-decadal variability no external forcing imprint is identified for this period. On the other hand, a general decrease in cyclone numbers (particularly for the Mediterranean) and cyclone related precipitation (e.g. north of 50_N over Europe) is identified for the XXI century. Finally, the authors discuss the possible relevance of thermodynamic vs dynamical processes for the identified trends / variability. The manuscript is well written, the methodologies and statistics are well applied, and the conclusions are largely sound. The consideration of such a long transient run is quite unusual, and the embedded discussion of natural vs anthropogenic forcing is quite interesting. Therefore, I believe the manuscript is a worthy contribution to Climates of the Past. Nevertheless, several minor aspects should be improved / better discussed before the paper is in acceptable form. Therefore, I recommend a minor revision according to the comments given below.

Minor Comments:

**1: lines 47-58: There is quite a lot of additional literature in this topic, so I understand the authors need to do a selection. However, I would recommend to include the two review papers of Ulbrich et al. (2009) and Feser et al. (2015), e.g. on line 48 and 52. If possible, a few more sentences on the different measures of cyclone activity and the regional differences would be excellent.**

We included the additional literature and extended the discussion.

**2: line 60: Please clearly state here that you mean that the low level meridional temperature gradients are reduced on average. On the upper troposphere, it is the opposite, as the strongest warming occurs in the tropical regions. Please find a suggestion below. This should also be stated more clearly other text passages. "The decrease of the projected low level meridional temperature gradient on average (due to strong high latitude near surface warming associated with polar amplification) implies a decrease of storm activity in the future, (: : :)<"**

We agree and clarified that the low level meridional temperature gradients are reduced on average. We clarified it here and in other parts of the manuscript.

**3: lines 161-163: While I understand the authors' idea to consider the 90th percentile of central pressure and cyclone depth as a proxy for windiness, I think it would have been easy to assign peak near surface wind speeds close to the cyclone core (e.g. Zappa et al., 2013) a more adequate measure of windiness associated with the cyclones. What has this not**

been done? Was the near-surface wind data not stored? Or was there another reason? This potential shortcoming makes a few statements in the manuscript (e.g. line 338-340) less robust and should at least be discussed as a potential shortcoming.

10 m wind data have not been stored. It is also a diagnostic variable which may suffer from some shortcomings, e.g., how the boundary layer is parameterized in the model. Above the boundary layer, the geostrophic approximation is reasonably fulfilled as we focus on the mid-latitudes. In particular, the cyclone depth measure takes advantage of the geostrophic approximation. Still, we clarified this. Concerning the lines 338-340 we do not see which shortcoming the reviewer refers to. Zappa et al. (2013) showed a slight decrease in extreme wind and an increase in precipitation using CMIP5 model simulation. We only use one model, so the only shortcoming we can see is that the multi model response deviates slightly from just using one model. We added a brief discussion in the conclusions.

**4: lines 185-199: It is a bit unusual that the (lower resolution) GCM has a higher cyclone frequency as the ERA-Interim dataset. While I tend to agree with the authors that this may be partially associated with an enhanced number of weak lows in the GCM, I wonder in how far the (bi-linear!!!) re-gridding of the ERA-Interim played a role here. What do the cyclone statistics with the original ERA-interim grid look like? Are the statistics more comparable if one only considers strong cyclones (e.g. exceeding a certain depth)?**

As expected the cyclone frequency increases slightly (roughly by 20 %) when using the original resolution of roughly 0.75 degrees. However, the main biases of the CESM simulation remain, namely the overestimation in the Hudson Bay and GIN Seas, so the interpolation does not change this result. We included this in the manuscript. Additionally, we calculated the cyclone frequency maps for strong cyclones, i.e., exceeding the threshold of 200 gpm in cyclone depth once in their lifetime. The threshold represents roughly the 10 % strongest cyclones. As expected the agreement is better for strong cyclones, but biases remain: the overestimation in the Baffin Bay and in the GIN Sea. We extended the discussion in the revised version.

**5: lines 301-310: Given that the main author has co-written a review paper on the NAO variability during the last millennium (Pinto and Raible, 2012), I wonder why so little is discussed about the link well established link between the NAO variability and cyclone variability over the Eastern North Atlantic and Europe (except for this text passage). In my opinion it would be pertinent to strengthen this statement and discuss a bit in how far the NAO variability in the simulation matches (or not) the cyclone variability for various parameters shown in Fig. 4, and in how far this agrees with NAO reconstructions. Even if the authors will surely explore this further in subsequent (and more regional) studies in the future, I suggest expanding the topic a bit here.**

Given the results of Pinto and Raible (2012) it is tricky to relate changes in cyclone characteristics linearly to the NAO. In the paper Raible et al. (2007) we found intensified cyclones during a long period (several decades) which was governed by a negative phase of the NAO. For year-to-year variations, a positive phase of the NAO is associated with intensified storms (e.g., Hurrell et al. (1997). So, a clear connection between NAO and cyclone characteristics seems to be time-scale dependent and is not necessarily expected. With the analysis presented in the paper we see that there is a dipole pattern which agrees in some aspects with the canonical NAO pattern but is shifted and the northern center is not barotropic (as the NAO concept suggests). This is the reason why we were cautious about the interpretation and called the pattern 'NAO-like'. To see whether the NAO has an impact on

cyclones one needs to perform a different analysis, i.e. going to more regional scales like a separation in Northern and Southern Europe, composite analysis with respect to the NAO index, etc. This is clearly beyond the scope of this paper, but is certainly of interest in future assessments of the simulation.

**6: lines 378-379: I suggest referring to Zappa et al (2013) here, which showed exactly this based on the CMIP5 model ensemble.**
Done.

**7: lines 381-401: The interesting thing here is that the increase in cyclone related precipitation is particularly clear north of 50_N (notably over Europe), while elsewhere reduced precipitation is often found, particularly at lower latitudes. Recent studies (e.g. Santos et al. 2016) have identified that there may be a "circulation independent" increase of precipitation north of _ 45_N over Western Europe and comparative drying around 35-45_N (cf. their Figure 9). This may imply that for the latter the increase of humidity is overcompensated by temperature (thus lower relative humidity) or hampered by increased subsidence. I think that present statement for the whole region regarding the Clausius-Clapeyron relationship is too general, and a more differentiated regional discussion would be quite interesting.**

Thank you for this comment. It seems that the reviewer has overlooked one important detail. In Fig. 7 we showed the mean precipitation and not the extreme cyclone-related precipitation. So, we can certainly compare this result with the one of Santos but for the discussion of the CC relationship, this might be a bit misleading. Realizing that this is a problem we tried to be more clear.
Concerning a more regional discussion of the CC relationship we implicitly have the regional scale included as we compare the extreme cyclone-related precipitation with the cyclone-related temperature. Again, we clarified this in the manuscript.

**11: line 424: Please add Ulbrich et al (2009) and Zappa et al. (2013) here.**
Done

**12: line 429: see discussion in #10, please enhance, maybe adding "at least north of 50_N" or similar.**
We think that adding north of 50N would be not supported by our analysis. See #10.

We thank the reviewer for his helpful comments and suggestions.

References (not exhaustive):
Feser F., et al. (2015). Storminess over the North Atlantic and Northwestern Europe: A review. QJRMS, 141, 350-382. doi:10.1002/qj.2364.
Pinto JG, Raible CC (2012) Past and recent changes in the North Atlantic Oscillation. WCC, 3, 79–90. doi:10.1002/wcc.150
Santos JA, (2016) Understanding climate change projections for precipitation over Western Europe with a weather typing approach. JGR-A, 121, 1170–1189. doi:10.1002/2015JD024399
Ulbrich U, et al (2009) Extra-tropical cyclones in the present and future climate: a review. TAAC, 96, 117–131. doi:10.1007/s00704-008-0083-8
Zappa G., et al. (2013) A multimodel assessment of future projections of North Atlantic and European extratropical cyclones in the CMIP5 climate models. JCLIM, 26, 5846– 5862. doi:10.1175/JCLI-D-12-00573.1, 2013.

**Anonymous Referee #3**

This manuscript investigates a long integration of the 1deg version of the CESM, from 850-2100 (with RCP8.5 forcings). The authors use 12-hourly data to track extratropical cyclones (ETCs) over the North Atlantic. Results are dominated by interannual-todecadal scale fluctuations of cumulative ETC metrics (count, intensity, precipitation) but no obvious external forcing signal is noted. After 2100, strong increases in ETC precipitation and decreases in ETC count are noted, with authors applying a regression analysis to demonstrate that these changes are mostly thermodynamic in nature, in line with previously published work. Some regional variations are also considered, particularly over the Mediterranean and Scandinavia.

In general, I feel the manuscript is clear and crisp, albeit not with overly novel conclusions. As a scientist who deals mostly with future storminess associated with climate change, I think this type of analysis is relevant to our understanding of climate models and the dynamics of the features themselves within the climate system over long time periods. Where I do have one concern is the results of the tracking algorithm, particularly with regards to CESM, that may be somewhat influencing the results. Before final publication, I feel these should be addressed by either retracking the storms or running a sensitivity analysis. Assuming the authors have a pipeline that performs the subsequent analysis in Figs. 4-8, this should be fairly trivial to undertake.

As one who has used CESM data in the past, if the authors are using the in-line 1000hPa geopotential (Z1000) as a variable (versus calculating Z1000 using the hybrid coefficients and topography) they are likely having issues with the fact that CESM will not automatically interpolate "below ground." Therefore, while the true Z1000 is likely negative over high-terrain areas (e.g., Greenland) the Z1000 reported from CESM is anomalously positive since the code will not go below the lowest model level (at least, according to my recollection). This is a quirk of the CESM in-line interpolation and is likely causing the issues (high cyclone count near high-terrain areas) seen in Fig. 2b since the "background" Z1000 field is biased very high. This can probably be rapidly verified by just comparing the time-mean Z1000 in both ERA-Interim (ERA-I) and CESM. The optimal correction for this would be to use some sort of offline solver with the 3-D Z field and Python/NCL/IDL/etc.

Actually we used the 3-D field of geopotential height and some NCL routines offline to interpolate the Z1000 field. The exact routines are:

```
;********************************************
; This script interpolates sigma to pressure coordinates
; in outputs the geopotential height (for all plevs chosen)
;********************************************
load "$NCARG_ROOT/lib/ncarg/nclscripts/csm/gsn_code.ncl"
load "$NCARG_ROOT/lib/ncarg/nclscripts/csm/gsn_csm.ncl"
;********************************************
begin
;***********************************************
; read in data
;***********************************************

;******only years
;******year has to be given with the function call: 'for a in { }; do ncl
;******year=... Interpolation_... ;done'

path1="/BPRD_trans/atm/hist/BPRD_trans.cam2.h1.$yy-01-01-00000.4.nc"

in = addfile(path1,"r")

;-----------------------------------------------------------------
```

```
; read needed variables from file
;---------------------------------------------------------------------

Z3 = in->Z3        ; select variable to be converted
P0mb = 1000.
hyam = in->hyam    ; get a coefficiants
hybm = in->hybm    ; get b coefficients
PS = in->PS        ; get pressure
TBOT = in->TS      ; get temperature at lowest layer (closest to surface)
dims = dimsizes(Z3)
nlevs= dims(1)
PHIS = Z3(:,nlevs-1,:,:)*9.81   ; get geopotential [m^2/s^2] at the bottom (lowest layer)

;---------------------------------------------------------------------
; define other arguments required by vinth2p
;---------------------------------------------------------------------

; type of interpolation: 1 = linear, 2 = log, 3 = loglog
interp = 2

; is extrapolation desired if data is outside the range of PS
; extrap = False
extrap = True

; A scalar integer indicating which variable to interpolate: 1 = temperature,
; -1 = geopotential height, 0 = all others.

varflg = -1

; create an array of desired pressure levels:
plevs =(/ 1000.0 /)

plevs!0 = "plevs"
plevs&plevs = plevs
plevs@long_name = "Pressure"
plevs@unit = "hPa"

intVar_PS = vinth2p_ecmwf(Z3,hyam,hybm,plevs,PS,interp,P0mb,1,extrap,varflg,TBOT,PHIS)

intVar_PS!0 = "time"
intVar_PS!1 = "lev"
intVar_PS!2 = "lat"
intVar_PS!3 = "lon"
intVar_PS&time = in->time
intVar_PS&lev = plevs
intVar_PS&lat = in->lat
intVar_PS&lon = in->lon
intVar_PS@units = "m"
intVar_PS@long_name = "Geopotential Height (above sea level)"

;system("echo saving")
;setfileoption("nc","Format","NetCDF4Classic")
;fileout=getenv("FZ")

fileout="BPRD_trans.cam2.h1.$yy-01-01.z1000.nc"

system("rm " + fileout)         ; remove any pre-existing file

fout=addfile(fileout,"c")

fout->Z3 = intVar_PS ; write into new file
system("echo new file for GPH")  ; print path and new file to screen as confirmation
```

Therefore, we think that a retracking is not necessary. To convince the reviewer we added an analysis of Z1000 and SLP in this point-to-point response:

Comparing the Z1000 with the SLP averaged for the period 1980-2009 a small positive difference over the center of Greenland is found (Fig R1, below). However, this difference between Z1000 and SLP does not affect the two regions where CESM overestimates cyclones (see manuscript Fig. 2), namely in the Hudson Bay and in between Iceland and Spitzbergen. We additionally made a visual test (movie of Z1000 field) and see that CESM simulates more

cyclones traveling to these two regions than ERA interim. So there are no stationary cyclones in these regions which may be a hint that the interpolation of Z1000 leads to some artificial cyclones. Note also the effect of the slightly coarser resolution of ERA interim (here in Fig. R1 1.5 degree) which leads to a less pronounced meridional pressure gradient in the North Atlantic. So some of the bias might be due to weak cyclones (we will give more arguments for this in the revised version), but certainly CESM also overestimates cyclones. We added the treatment of the vertical interpolation in the revised version.

[Figure]

*Fig.R1: Comparison of mean SLP and Z1000 field for winter DJF for the period 1980 to 2009: (top) CESM, (bottom) ERA interim.*

In this vein, it is not clear why the authors are not tracking on sea level pressure (PSL), which is essentially a prognostic quantity in most climate models (technically PS is prognostic, but the correction to PSL primarily uses other prognostic variables like T and the surface topography field). PSL is a much more widely-used quantity when evaluating climate models and would likely alleviate the issues.

As seen above there is no strong difference between PSL and Z1000 so that we think it is not necessary to redo the analysis with PSL. Also from a dynamical point of view the geopotential height is more relevant than the PSL and there is a direct relation to the geostrophic wind. In most publications, which apply the cyclone detection and tracking method of Blender et al. (1997), Z1000 is used as input rather than sea level pressure. For consistency, we would like to stay with Z1000.

The fact that CESM simulates far more cyclones than ERA-I is therefore questionable. While there are certainly some differences in effective resolution, etc. of the datasets, a factor of almost 2x (Line 202) in the total number of storms between CESM and ERA-I seems quite high at first blush. The authors hypothesize this is due to weak C2 storms, but that is not clear to me from Fig. 3. For example, the SLP distribution shows more weak storms for CESM, but also more strong ones. Having a smaller radius distribution is also not necessarily indicative of weaker storms, as aspects of the model configuration such as numerical diffusion and how grids are interpolated may contribute to differences here. This is somewhat hinted at in Figs 3c-d. As an additional example, one could make an argument that 1deg ETCs would be "smaller" than 4deg ETCs, but 1deg ETCs *should* be more intense based on being better resolved.

We agree with the reviewer that the number of cyclones are strongly overestimated in CESM. One reason is that CESM shows more short-lived cyclones. When comparing the cumulative

cyclone presence, the difference is already reduced (CESM: 22993; ERA interim: 15590). Though, CESM still overestimates cyclones. We further agree with the reviewer that some of the argumentation explaining this difference was not clear and Fig. 3 does not always support the line of evidence given. Therefore, we reformulated the entire paragraph. We also performed additional analysis, e.g., conditional distributions for short-lived cyclones which show that they can explain some of the difference in the number of cyclones. Also we find that short-lived cyclones have a smaller radius and are less intense (based on central pressure and cyclone depth). We also redid part of the analysis with the ERA interim in 0.75-degree resolution. As expected, we find more cyclones (roughly 20 %) in the higher resolution compared to the lower one. Also the mean radius is increased by roughly 10 % in the high resolution ERA interim. However, the biases discussed above and in the manuscript still remain present when comparing CESM to the high resolution ERA interim results.

I would like the authors to consider "retracking" the storms if PSL is available. They could easily modify their algorithm to search for prognostic deficits in PSL as in other trackers within the IMILAST project (of which the lead authors of this manuscript already contributed to). If that is not available, I would like the authors to try and evaluate whether or not the issues of additional ETCs tracked in CESM are related to the Z1000 issue noted above. One option would be to run CESM for a short period (perhaps a few decades) and compare the results of using the inline Z1000 with PSL or a more accurately diagnosed Z1000.

As seen by the analysis above, the mean PSL and Z1000 pattern are very similar. The suggested use of PSL in the detection and tracking method of Blender et al. (1997) induces substantial changes. It would involve testing and adjusting several parameters of the method and given the fact that the mean pattern in Fig. R1 look very similar, we think it is not necessary to "retrack" the cyclones.

Minor comments:

Line 122: "So called" is too colloquial, would just say "this is the 1deg version of the model used in CMIP-class experiments" or thereabouts.

Done

Line 123-124: Would include a sentence or two about the subgrid physics package used in this version of CESM (in the atmospheric model) since that would have the largest impact on the results here, particularly thermodynamic ones.

The main changes of CAM (atm. model) are described in the Neale et al. (2013). We added a brief description in this section.

Line 245: Are there changes in mean storm-track, basin-wide surface pressure, etc. that may be relevant here?

The SLP can be affected by a basin wide surface pressure trend (as it is part of it), but the cyclone depth (as a relative measure) is not sensitive to such a change. As results from both measure agree with each other, we do not expect an influence of basin-wide surface pressure trend on this result.

Line 262: 4deg models certainly underresolve synoptic scale features, which ETCs are.

We clarified this.

Line 299: Is this a basin-wide metric? I question a bit about correlating the spatial pattern with basin-wide metrics as then I'd expect large scale North Atlantic patterns that control ETCs to dominate this analysis (e.g., the NAO).

Yes, basin-wide metrics are used to get a first more general impression. We expected to see NAO-like patterns which is (to some extent) true for the cyclone depth metric but not for the cyclone-related precipitation.

Line 312: The spatial field remains quite noisy, I would be a bit careful about being too conclusive since, even with a multi-century simulation, I'm not sure we can be completely confident very small (O(10deg)) spatial patterns are tremendously significant in a model whose effective resolution is probably _6deg (see Skamarock 2004 for discussion of effective resolution in numerical models).

It is expected that the Figs. 5c and d are noisier than a and b as we assess cyclone frequency and precipitation. We are aware that one should only interpret significant areas of a certain larger extent so we modified the description of Fig. 5c,d accordingly. We also realized that not always the region of the significant change was mentioned. This is also clarified.

Line 321: This reads as a bit "hand-wavy;" I'd formalize and clean up the text a bit.

We tried to reformulate this part.

Line 332: These "barotropic pressure structures" could be underresolved warm core storms (e.g., tropical cyclones) moving to mid-to-high latitudes. 1deg models are capable of starting to simulate these features, albeit far weaker than what is observed (e.g., Wehner et al., 2014, Walsh et al., 2015).

We realized that this was not clear. The barotropic structure we refer to is over Central Europe. So, we have high pressure when the cyclone related precipitation index is enhanced. As most of the correlation coefficient in the Z1000 and the Z500 field are not significant we removed this part. Note that hurricanes cannot play a major role as we focus on season DJF.

We thank the reviewer for his helpful comments and suggestions.

We tried to be more precise in the revised version of the manuscript and extended some of the physical explanations.

We thank the reviewer for his helpful comments and suggestions.

[Figure]

Fig. R2: Wavelet (left and middle column) and crosswavelet (right) spectra. The relative phase relationship is shown as arrows in the right panels (with in-phase pointing right, anti-phase pointing left, and solar forcing leading the cyclone charcteristic by 90° pointing straight down). The method is taken from Grinsted et al. (Nonlinear Processes in Geophysics, 2004, 11: 561–566).

[revised manuscript text omitted]